# Impact of Situational Environmental Education on Tourist Behavior—A Case Study of Water Culture Ecological Park in China

**DOI:** 10.3390/ijerph191811388

**Published:** 2022-09-09

**Authors:** Jinming Wang, Jialu Dai, Bart Julien Dewancker, Weijun Gao, Zaiqiang Liu, Yue Zhou

**Affiliations:** 1Faculty of Environmental Engineering, The University of Kitakyushu, Kitakyushu 808-0135, Japan; 2Faculty of Environmental Systems, The University of Kitakyushu, Kitakyushu 808-0135, Japan

**Keywords:** situational environmental education, tourists’ responsible environmental behavior, planned behavior model, mental models

## Abstract

With the increasing number of travelling people, the behavior of tourists is having an increasing impact on the environment. Situational environmental education will influence the tourists’ responsible environmental behavior, which positively or negatively affects the environment. The purpose of this study is to explore the impact of situational environmental education on tourists’ responsible environmental behavior through a field study of Changchun Water Culture Ecological Park, combined with a survey and Zaltman metaphor elicitation technique (ZMET) interview method. There are 527 questionnaires, 89 pre-questionnaires, and 15 interview records collected. The results showed that: (1) All interviewees were impressed with the situational environmental education in the park. It can be concluded that the situational environmental education is easily accepted. The reason may be that, among the theme park users, 42.69% were aged 21–30 year’s old, and 62.8% of the population have a college degree or above. (2) The standardized path coefficient of situational environmental education in tourist destinations for tourists’ behavioral intention is 0.74, and the standardized path coefficient for responsible environmental behavior is 0.78, which is much higher than the standard value of 0.4. Therefore, situational environmental education has positive influences on the tourists’ behavioral intention and responsible environmental behavior. (3) The sensitive analysis the tourists’ behavioral intention has a positive relationship with attitudes toward environmental behaviors (0.66), subjective norm (0.53), and perceptual behavior control (0.52). The results of this study can provide a scientific basis for the planning and design of urban parks.

## 1. Introduction

### 1.1. Background

According to the World Tourism Council (WTTC), tourism contributed USD 8.9 trillion—or 10.3%—to global gross domestic product (GDP) in 2019 [1]. According to the law of international development, the epoch of popular tourism has arrived, and tourism has now become a necessity for thriving economies [2].

Even though increasing numbers of tourists bring huge economic benefits, they create a lot of pressure on the local environment [3]. Environmental problems, such as ecological damage and pollution, have become increasingly prominent in tourist destinations. Most tourists have minimal sense of restraint, environmental protection, and self-control, leading to uncivilized behaviors [4,5,6]. Such behaviors cause ecological environment problems in tourist destinations and are a challenge to the management and protection of the ecological environment of these destinations [7], leading some countries to explore more effective tourism management strategies [8].

Environmental education can be used as an important tool for solving environmental problems as it strives to achieve the goals of environmental protection and ecological protection [9]. Environmental education aims not only to influence individuals’ internal characterizations and understandings of the world, but ultimately to motivate people from within to adopt appropriate real-life behaviors [10]. In fact, education is considered an essential requirement if we want to successfully promote sustainable development [11].

Understanding the intrinsic motivations exhibited by behavior is key to reducing adverse human impacts on the planet [12], as extrinsic motivations (e.g., incentives, punishments) are often opposed and have only temporary effects [13]. Therefore, fostering intrinsic motivation for ecological behavior is essential to truly mitigate anthropogenic environmental problems [12]. Most of the existing studies tend to focus on the environmental education people received during childhood [14,15], while ignoring the environmental education that tourists receive unconsciously through the environment during tourism.

Tourism destinations are places where tourists engage in environmental behaviors [4,5]. ‘Responsible environmental behavior’ is a collective term for the behavior of tourists to engage in and address environmental issues. According to the “Hines Model of Environmental Behavior” constructed by Hines in 1985, responsible environmental behavior comes from three levels: (1) the expectation level: including attitudes, perceptions of control and personal responsibility; (2) the action skill level: including knowledge of action strategies and problem awareness; and (3) the action intention level [16,17,18]. It is not only related to the individual, but also to the whole surrounding environment, which is rooted in social values/norms and influenced by institutions, programs/policies, scenic environment, and many other aspects. Here, travel behavior differs from many other forms of behavior. Bagozzi pointed out that important situational factors may influence the relationship between intention and behavior [19]. These situational factors are existence and transience and are perceived by the tourist during the journey [4]. It is worth noting that situational factors have attracted increasing attention in studies on tourism, especially situational planning, tourism, marketing, and consumption [20,21]. However, research on situational factors of tourism destinations is limited [22].

Situational environmental education is an educational model based on naturalness and conservation [23]. Objective situational environmental knowledge education is the information transmission of tourist destinations to tourists, emphasizing the type and nature of information [24,25].

Subjective situational environmental education can be seen to some extent as an education that promotes tourists’ self-perception of environmental knowledge; that is, allowing tourists not only to receive objective environmental knowledge, but also to perceive themselves as having corresponding environmental knowledge [26,27].

Most previous studies on tourists’ behavior toward the environment have focused on tourists’ own psychological factors [18,28,29]. Several studies have paid attention to the environmental background situational factors of tourist destinations, which directly influence the behavior of tourists [5]. However, this also ignores the environmental education that tourists unconsciously receive during tourism. Situational environmental education, as an important situational factor, has a more direct impact on tourists’ responsible environmental behavior. Therefore, it is particularly important to explore the mechanisms of the impact of situational environmental education on tourists in tourism destinations and the appropriate entry points for intervention.

Mental models are the lens through which people view the whole world [30]. They comprise people’s experiences, beliefs, learning, values, and understanding of how this world works [31,32]. The mental models represent schemas of the structure of an individual’s perception of situations, experiences, and issues [33]. These perceptual structures also consist of attitudes, emotions, images, behaviors, signs, memories of past events, targets, personal values, desires for expected events, and perceptual images [34]. Mental models have been proved a very significant factor in the process of problem-solving [35].

These images indicate that metaphors of cause-and-effect relationships are outlined in the mind. Many scholars believe that daily reasoning is based on metaphors [36]. Using images in such interviews has been proven to inspire deeper meanings and provide useful insights into the emotional factors and intangible thoughts of the interviewee [37,38], different from what traditional face-to-face interviews conducted in previous studies found [39,40]. Thus, a complete understanding of ideas, experiences, and values can be acquired from using images, through the representation of the human unconscious state [41].

To understand the unconscious experience expressed by the interviewees, it is necessary to use an exploratory method that can elevate the experience from unconscious to clearly expressible. The means-end chain (MEC) method and the related stepped interview technology are used to apply the Zaltman metaphor elicitation technique (ZMET) that explores the deeper thoughts in the mind of the interviewee [42,43]. Additionally, grounded theory methods are applied for collecting data, because they enable the finding of rich and sophisticated data and help in the determination of the main topics by coding and grouping the interviewees’ concepts to obtain their perceptual structure [44,45].

In order to understand the unconscious experiences expressed by respondents, it is necessary to use an exploratory approach that can elevate the experience from the unconscious to clearly articulable. The means-end chain (MEC) method and related stepwise interview techniques are used to apply the Zaltman metaphor elicitation technique (ZMET), which explores the deeper thoughts in the respondent’s mind [42,43]. At the same time, basic theory methods are applied to collect data, as they allow finding rich and complex data and help to identify the main topics [44,45], by coding and grouping respondents’ concepts to obtain their perceptual structure.

This research investigates the effects of situational environmental education on tourists’ responsible environmental behavior and avoids the shortcomings of a single questionnaire method through methods such as ZMET and MEC. In this process, situational environmental education is added as a covariate to the Theory of planned behavior (meta-analytic integrated model reference). The image data generated by the respondents are used to derive the conclusions of the analysis [36]. In this model, tourists need to truly generate responsible environmental behavior and need to develop intrinsic motivation through a sense of connection to nature in conjunction with situational environmental education. Environmental education provided by themed tourism destinations. Tourists receive nature-based environmental education in or close to nature at theme destinations, learning about the environment while promoting a connection between tourists and nature.

### 1.2. Conceptual Framework

#### 1.2.1. ZMET

The Zaltman metaphor elicitation technique is a qualitative research method. It allows for an in-depth exploration and study of the intrinsic needs and ideas of tourists [46]. More than 80% of human communication does not come from words or language [36]. Survey methods in the traditional sense—such as questionnaires, focus groups, face-to-face interviews, etc.—basically expect users to express their ideas in words or language. Only about 5% of user awareness can be obtained in this way, while 95% of awareness is not available [47]. ZMET combines non-textual language (images) and textual language (in-depth interviews). It starts with tourists as the main body, chooses pictures as a communication medium, and brings out the potential thoughts and feelings in the tourists’ mind through the metaphorical function of visual symbols in the pictures. Finally, a consensus map was created to show the results of the perception of the specific problem [36]. Studies have shown that tourists unconsciously lie when answering questionnaires because of surrounding situational factors, overlap effects, or overlimit effects, which can lead to false conclusions [48]. Therefore, it is very important to choose ZMET to guide visitors to express the factors in their mind that really affect their REB.

#### 1.2.2. MEC

Respondents may need to explore ways to elevate experiences from the unconscious to a level where they can be clearly articulated. One such technique used in ZMET is the MEC method [42] and its related stepped interview technique [43]. In this regard, the grounded theory approach is used for data collection, allowing the discovery of rich and complex data and the identification of key themes [49]. These can be coded and grouped into concepts and further categorized in order to map cognitive structures from the respondents’ mental models.

In its original form, the MEC approach describes consumer behavior as based on the attributes consumers perceive in a consumer product, the consequences associated with those attributes, and the consumer’s perception of how those consequences function to satisfy a desired end state or value. The usefulness of MEC is to identify the hierarchy from attributes to outcomes to values. In this way, it allows for the identification of the mental representations of values that drive decision making, and how attributes of a product can lead to the satisfaction of certain values through related consequences. Although derived from consumer research, MEC has been found to be a useful method for understanding decision making in other contexts, such as farmers’ decisions regarding animal welfare [50], soil management [51], environmental health [52], pesticide use [53], and environmental attitudes and behaviors [54].

#### 1.2.3. NAT

In comparison to TPB, which is essentially a general theory of behavior, NAT is originally developed specifically for one type of behavior—that is, altruistic and helping behavior. The basic assumption of NAT’s theory is that if people feel morally obligated to help others in a given situation, that behavior is known as an activated personal norm. After Thøgersen’s research [55], many researchers have applied NAT to explain behavior that is important to the environment and have achieved promising results showing that pro-environmental behavior is actually influenced by NAT variables [56,57]. In comparison to TPB, NAT focuses strongly on the moral drivers of pro-environmental behavior and ignores the non-moral motivations that TPB would be captured.

#### 1.2.4. TPB

TPB was proposed by Ajzen in the early 1990s as a general model of intentional behavior [58]. The center hypothesis is that the behavior is directly determined by the intention to perform this behavior. This intention in turn depends on attitudes toward the behavior, subjective norms associated with the behavior, and perceived behavioral control [59]. Attitudes are the sum of all behavioral beliefs about what is activated in a given situation. Beliefs are an assessment of the expectation that exhibiting a certain behavior will lead to a certain outcome, the likelihood that this will occur, and the extent to which it will be beneficial to that outcome. Thus, attitude is a general measure of favorability to behavioral choices. Subjective norms are the perceived expectations of relevant other people which behavioral alternative should be performed (in other words the social pressure) and the willingness to comply with that expectation. Finally, perceived behavioral control is a measure of the degree to which people have the opportunity and ability to execute a certain behavioral choice. It can therefore be used as a proxy measure for actual control conditions and thus directly predict the likelihood of behavior occurring [42,60].

According to TPB, people will perform behaviors with positive environmental outcomes if they hold positive attitudes toward them, if others expect them to behave in this way and support them in doing so, and if they believe they can achieve their intentions. It is important to recognize that all three structures are subjective representations, meaning that perceived control is not necessarily the same as the objective or actual control people have, or that subjective norms do not necessarily reflect the true expectations of others. Perceived behavioral control can have an additional direct effect on behavior under certain conditions, such as when conditions change, before the behavior is performed. In the field of ecological behavior, TPB has been shown to help explain travel mode choice [59], recycling behavior [61], water conservation [62], and ecological consumer behavior [63].

Compared to the value-belief-norm-theory, which assumes that behavior is directly determined by personal norms based on NAT, and the NAT theory, which was originally developed specifically for altruistic and helping behavior, theory of planned behavior, in which behavior is directly determined by the intention to perform this behavior, is more consistent with the model of responsible behavior for tourists. Therefore, this paper chooses the theory of planned behavior as the underlying framework.

#### 1.2.5. Kelly Repertory Grid Technique

Kelly repertory grid technology is an important mixed-methods analysis technique that began in the field of psychology and has been widely used in the medical, consulting, and educational fields. In the field of education, it is often used to explore the cognitive structure of cases [64] and the change and growth of cognitive structures [65].

## 2. Research Hypotheses

### 2.1. Perception of Environmental Behavior

Perceptions of environmental behavior concern the pros and cons of the environmental behavior of tourists, and it is mostly affected by behavior belief. Behavioral belief is a subjective evaluation of the possible consequences of a behavior when the individual performs that behavior [4,5]. According to the TPB analytical framework, perception of environmental behavior is a valid predictor that impact the REBI of the tourists. Related studies have affirmed that tourists’ perception of environmental behavior can predict their REBI [2,66,67]. Based on these findings and our analyses of them, we proposed our first hypothesis:

**Hypothesis** **1** **(H1):**
*Tourists’ attitudes toward environmental behavior have a positive impact on their responsible environmental behavioral intentions (REBI).*


### 2.2. Subjective Norms

Subjective norms concern the social pressure on tourists about generating or not generating some environmental behaviors and can be influenced by motivation (for compliance) and normative beliefs. Normative beliefs are beliefs imposed by a specific person or group. Compliance motivation means whether it is appropriate to behave in a specific way under the influence of the belief. Related studies affirm that tourists’ subjective norms can predict their REBI [68,69]. Therefore, under the analysis framework of TPB, SNs are effective predictor variables that affect the REBI of tourists. Thus, we propose our second hypothesis:

**Hypothesis** **2** **(H2):**
*Tourists’ subjective norms have a positive impact on their responsible environmental behavioral intentions (REBI).*


### 2.3. Perceptual Behavior Control

Perceptual behavior control (PBC) is the visitor’s judgment of whether a behavior is difficult to occur [65] and can reflect experiences, anticipated difficulties, and barriers. Controlling beliefs, or factors that may promote or hinder certain behaviors; and self-power, or one’s estimate of whether one can successfully adhere to their behavior under current conditions, can impact tourists’ PBC. Relevant studies affirm that tourists’ PBC can predict tourists’ REBI and REB [2,67], which gives us our third and fourth hypotheses:

**Hypothesis** **3** **(H3):**
*Tourists’ perceptual behavior control has a positive impact on the responsible environmental behavioral intentions (REBI) of the tourists.*


**Hypothesis** **4** **(H4):**
*Tourists’ perceptual behavior control has a positive impact on the responsible environmental behaviors (REB) of the tourists.*


### 2.4. Environmental Behavior Intention

Environmental behavior intention refers to the subjective probability of tourists performing a certain environmental behavior. As a direct predictor of REB, environmental behavior intention has a remarkable effect on the relationship between SNs, PCB, environmental behavior perception, and REB [4,5]. Tourists’ REBI can directly predict their environmental behaviors [41], which brings us to the fifth hypothesis:

**Hypothesis** **5** **(H5):**
*Tourists’ responsible environmental behavioral intentions (REBI) has a positive impact on their responsible environmental behaviors (REB).*


Linda Steg, Wang et al., argued that the above hypotheses are correct. The authors argue these hypotheses again, which not only authenticates the previous studies but also validates the degree of coupling of the EE-TPB model [4,5,70].

### 2.5. Impact of Situational Environmental Education

Tourists react strongly to the situational environmental education of their destination. When sustainability education is implemented, tourists consider factors, such as environmental integrity and cultural diversity before their environmental behavior takes over. For example, when climate change education is conducted, tourists think of smog, environmental warming, and rain and snow. Similarly, after garden-based learning is implemented, tourists’ attitudes towards the environment can change. Additionally, the development of outdoor education allows tourists to appreciate nature more and learn and cultivate pro-environmental awareness and behaviors. When tourists receive situational environmental education information, their REBI and REB are affected, giving us the final hypotheses:

**Hypothesis** **6** **(H6):**
*Situational Environmental education about the destination has a positive effect on the responsible environmental behavioral intentions (REBI) of the tourists.*


**Hypothesis** **7** **(H7):**
*Situational Environmental education about the destination has a positive impact on the responsible environmental behaviors (REB) of the tourists.*


## 3. Data Sources and Research Methods

### 3.1. Overview of the Study Area

Changchun Water Culture Ecological Park is located in Changchun City, Jilin Province, China, covering a total area of 50,000 square meters. (Figure 1a,b) The daily guest visits can reach more than 30,000 people. It is a key urban construction project in Changchun City which has fully preserved water purification equipment from different ages. Through the combination of animation and traditional water purification equipment, some of which is presented as artwork, visitors are informed scientifically about water purification (inquiry-based science or IBS) and water supply and drainage processes through personal experience (experiential education or ExE). (Figure 1c,d) The landscape design highlights the city system, comprising a variety of environmental protection systems, including a slow-moving system, primitive animal and plant ecosystem, and water ecological self-purification system. The forest corridor that runs through the park minimizes damage to the original vegetation system, while bringing people a comfortable aerobic stroll experience (outdoor education or OE). Several concrete sedimentation and clean water tanks have been retained in the park. This method of preserving and strengthening the personal characteristics of the site minimizes damage to the environment, while allowing tourists to gain maximum knowledge of the water plant (garden-based learning or GBL). The project uses ecological green space as the carrier, allowing tourists to unconsciously achieve the purpose of situational environmental education.

### 3.2. Research Method

#### 3.2.1. ZMET Interview Method

We identified visitors with high involvement in Changchun Water Culture Ecological Park through Revised Personal Involvement Inventory (RPII). The mental model of this part of tourists can represent the mental model of most tourists [71]. In-depth interviews were conducted through ZMET technology. The implementation process of the metaphor extraction technique follows the 10 steps of metaphor extraction in sequence. The interview data were summarized and organized, including conceptual extraction, integration and classification, and hierarchical classification of the factors mentioned by visitors using MEC. After building the mental map of tourists, by collecting and organizing them to build a value chart, we can identify factors that influence the responsible environmental behavior of tourists.

The study combines ZMET and MEC to construct a mental model of tourists. Interviews are conducted for highly involved respondents. The model mradually moves from concrete attributes to abstract attributes and eventually escalates to the most abstract personal values, establishing the correlation structure of the mental model. In this way, it is possible to gain further insight into the meanings and ideas behind the concepts, as well as gain insight into the deeper connections between thoughts and feelings in the respondents’ psychological structure. This leads to the deep-level factor that affects the responsible environmental behavior of tourists.

ZMET can present the user’s mental map, sensory and perceptual images, summary images of the destination, small texts, and other rich meanings that can give us an understanding of the user’s mental process. The whole process is shown in Figure 2.

#### 3.2.2. EE-TPB Model

In this study, we applied situational environmental education as a variable and added it to TPB, thus constructing an analysis model of situational environmental education-tourists’ REB. The five former variables of SNs, environmental behaviors, environmental behavior intention, PCB, and situational environmental education are regarded as situational factors of tourism destinations. Figure 3 portrays the relationship between situational environmental education and TPB.

### 3.3. Respondents and Process

#### 3.3.1. Personal Involvement Measure

For selection of interviewees, we first considered relevance (degree of involvement). Tourists with high involvement are likely to have more knowledge and experience of the destination, and thus, their consensus models would be more effective. To ensure the representativeness of the interviewees, we analyzed the answers provided by pre-interviewees and interviewees, and used the personal involvement measurement of the pre-interviewees as the filtering condition for the final interviewees.

The respondents’ sensitivity to language and attention might decline during the test. Therefore, without affecting the reliability and validity of the inventory, 20 pairs of variables were condensed into 10 pairs, and the Revised Personal Involvement Inventory (RPII) was proposed. Each variable had seven scales corresponding to seven points; therefore, the total score ranged from 10 points to 70 points. The higher the score, the higher the degree of involvement in the topic: 10–35 points (low involvement), 36–55 points (medium level of involvement), and 56–70 points (high level of involvement) [72].

The pre-questionnaire began on 16 September 2021. We invited 89 pre-interviewees through a combination of follow-up and the snowball method. As shown in Table 1, 15 interviewees were highly involved and met the study requirement. Their RPII scores were over 56 points, indicating their deep involved in the destination, and met the high-relevance requirement of our study.

#### 3.3.2. Guided Interviews

Interviews began in November 2021. Before the interview, interviewees were asked to collect pictures that express their feelings about the destination landscape. For each interviewee, we made an appointment 7–10 days in advance through e-mail and telephone and stated the theme and purpose of our study clearly during the interview. Images were collected by us from newspapers, magazines, and books, or by downloading photographs from the internet.

During the interview, ZMET was used as the main method, the ladder method [73] was used for in-depth interviews, and the Kelly grid method was used to find out the factors that influence the interviewees’ REB. The total number of extracted ideas was divided into common constructs and relative constructs, based on the “Three Most” principle—most of the time, most of the people, and most common ideas. The specific calculation method involved common constructs (more than one-third of the interviewees mentioned them twice or more) and relative constructs (more than a quarter of the interviewees mentioned them twice or more). The ZMET research method attaches more importance to the relationship between multiple constructs than to a single construct; therefore, even when a certain construct was mentioned by all respondents (construct 1), if each interviewee had no consensus on the correlation between construct 1 and other constructs, then construct 1 was considered unimportant.

Guided interviews mainly focused on collecting specific image symbols, which were from the images collected by the interviewees. These symbols included contexts, actions, expressions, occasions, objects, characters, objects, and hidden meanings. We extracted the content that the interviewee wanted to express, which are their true thoughts in their consciousness. Then, we summarized the extracted constructs and the connections between them. Finally, the personal consensus map of the 15 interviewees was created. Table 2 below gives the details about the ZMET process applied.

### 3.4. Situational Environmental Education-TPB Questionnaire Design

Combining previous research, a questionnaire was designed to study the predictors of the tourists’ REB. Attitudes to PCB, SNs, environmental behaviors, REBI, and situational environmental education used a five-point Likert type scale. Table 3 summarizes the different constructs and scale items applied in our study.

## 4. Analytical Methodology

### 4.1. Data Acquisition

The investigation was conducted from 16 September to 14 November 2020. The author hired some college students to distribute questionnaires in the park, and briefly introduced the project to some of the park staff and the questionnaire distributors in advance. A total of 543 questionnaires were sent out, and 540 were recovered. The questionnaire set a lying coefficient the question “I think it is waste of time to protect the environment” was used to eliminate 18 invalid questionnaires. Finally, 527 valid questionnaires were obtained. The recovery rate and the effective rate of investigation were 99.44% and 97.01%, respectively. In our study, one person entered the data into SPSS 22.0, a second checked the data, and a third performed random checks to ensure accuracy.

All actions are consistent with the Declaration of Helsinki [74]:All persons tested completed the questionnaire voluntarily and consented to the use of the data.Relevant events such as the purpose of the experiment were stated in all questionnaires.All questionnaires were filled out anonymously.Psychological interview techniques (ZMET, MEC, etc.) have been licensed for use in this study.All respondents signed an informed consent form.

### 4.2. Analytical Processing

Structural equation modeling statistical methods were applied to test the suppositional relationships between the situational environmental education, the TBM model, and its variables. We used confirmatory factor analysis (CFA) to evaluate the internal structure of the integrated model in AMOS 21.0 (SPSS Inc., IBM, Chicago, IL, USA), and SPSS 25.0 software (SPSS Inc., IBM) for its reliability and validity.

The interview structure that followed the eight ZEMT steps ensured that the same interview procedures were performed for all respondents. The hierarchical value map (HVM) summarized the complicated interview material into an easily accessible visual representation, which indicated the factors that visually influence the tourists’ REB. A critical value of 7 was added for the aggregated HVM, which corresponded to 46.7% of all A-R-V connections, which meant that at least seven constructs were elicited from the overall set of A-R-V connections established from interview materials. These derived structures were further grouped under the overall content code and were represented by different boxes in the HVM [39]. In HVM construction, the thickness of the lines indicates the strength of the relationships between them the thicker the line, the more often this relationship is derived. The “Sub %” in the HVM box indicates the percentage of interviewees who initiated the construction, and the “nr” indicates the number of times that the construction was initiated.

## 5. Results

### 5.1. Analysis of Metaphor Extraction Conclusion

In this study, the methodological advantages of the ZMET were particularly reflected in storytelling and enlightening, which deliver the richest descriptions to discover the interviewees’ understanding and relationship between consequences and attributes, and eventually related individual values. In other studies, other steps offered value by confirming the findings, instead of providing supplementary information [52]. In the process of analyzing materials from the 15 ZMET interviews, a sum of 42 steps were determined. There were 166 structures in total, and 57 connections were formed.

HVM indicated that when tourists visited the Changchun Water Culture Ecological Park, the various forms of education had a profound impact on their behavior or motivation. Fifteen tourists remembered the water purification process very well, and several specified that, although they could not remember the environmental knowledge involved, they were motivated to protect the environment. Thus, both the situational and individual factors of the tourist destination had a corresponding impact on the tourists’ behavior. Table 4 depicts the HMV map and Figure 4 presents the consensus map developed in our study.

### 5.2. Situational Environmental Education-Theory of Planned Behavior (TPB) Model Result Analysis

#### 5.2.1. Descriptive Statistical Analysis of the Sample

As shown in Table 5, the 527 tourists who participated in the survey portrayed the following characteristics:(1)The number of visitors under 20 and over 60 was relatively small; mainly young and middle-aged people visited the park.(2)The educational level of visitors was generally high, and the interviewees were mainly university students.(3)Most of the tourists were visiting the park for the first time.

**Table 5 ijerph-19-11388-t005:** Demographic data of the visitors of Changchun Water Culture Ecological Park interviewed for our study.

Characteristics	Category	Quantity	Percent (%)
Gender	Male	275	52.18
	Female	252	47.82
Age	≤10	4	0.76
	11–20	12	2.28
	21–30	225	42.69
	31–40	122	23.15
	41–50	96	18.22
	51–60	62	11.76
	≥60	6	1.14
Education	Elementary school and below	10	1.9
	High school	186	35.29
	College	265	50.28
	Master’s degree and above	66	12.52
Occupation	Public official	39	7.4
	Business personnel	28	5.31
	Mechanics/workers	104	19.73
	Waiters/salespersons	18	3.42
	Company staff	156	29.6
	Student	81	15.37
	Retired people	41	7.78
	Others	60	11.39
Place of residence	In Changchun	337	63.95
	Outside Changchun	190	36.05
Frequency of visiting the Changchun Water Culture Ecological Park	One time	399	75.71
	Two times	76	14.42
	Three times or above	52	9.87

#### 5.2.2. Confirmatory Factor Analysis

Composite reliability (CR) was applied to measure the degree of various internal relationships, that is, reliability. Cronbach’s Alpha was applied to evaluate the questionnaires’ reliability and consistency, that is, validity. As shown in Table 6, the alpha values of the measurement scales were between 0.739 and 0.914, both higher than the standard of 0.7. Thus, the data itself had interior reliability and consistency [75,76]. The composite reliability values of six latent variables ranged from 0.603 to 0.946, higher than the standard of 0.6, suggesting high consistency [77].

Attitudes toward environmental behaviors (ATEB), responsible environmental behaviors (REB), responsible environmental behavioral intentions (REBI), situational environmental education (EE), perceptual behavior control (PBC), subjective norm (SN), composite reliability (CR), average variance (AVE), and standard deviation (SD).

We used the standardized factor load and the extracted average variance (AVE) (Table 6) to examine the validity of the measurement model convergence. The 18 standard factors loading variables observed were between 0.693 and 0.935, above the standard of 0.5, suggesting that each observed variable had high explanatory power for its latent variables [78]. All the latent variables of AVE values were between 0.819 and 0.909, higher than the standard of 0.5. This indicated that the mean explanatory power of the total items in the questionnaire was sufficient [77]. Discriminant validity tests were performed by comparing the square root of the average of latent variables with the correlation coefficient between the latent variables.

The sample size was 527 over 50, so we used the Jarque-Bera test. As shown in Table 7, the absolute value of kurtosis was less than 10, the absolute value of skewness was less than 3, and the *p*-values were all greater than 0.005, so we believe that ATEB, SN, PBC, EE, REBI, and REB were all consistent with normal distribution.

As shown in Table 8, all the latent variables’ square roots were greater than their correlation coefficients with other latent variables, showing discriminative validity. R^2^ was greater than 0.5 and the Pearson correlation coefficients were less than 0.01, so the fit of the linear regression between the variables was excellent and the correlation between the variables was high.

#### 5.2.3. Structural Equation Modeling

Table 9 and Figure 5 show the results of model fitting applied in our study, which were:(1)Standardized path coefficient from ATEB to REBI was 0.660; thus, ATEB and REBI were highly positively correlated, verifying H1.(2)Standardized path coefficient from SN to REBI was 0.530, indicating that SN and REBI were highly positively correlated, and verifying H2.(3)Standardized path coefficient from PBC to REBI was 0.514 and to REB was 0.525, indicating high correlation between PBC, and REBI and REB. Thus, H3 and H4 were verified.(4)Standardized path coefficient from REBI to REB was 0.804; thus, PBC was highly correlated with REBI and REB, and H5 was verified.

Therefore, ATEB had a positive impact on the tourists’ REBI, SN also affected their REBI positively, the tourists’ PBC positively influenced their REBI and REB, and the visitors’ REBI positively influenced their REB as well.

(5)Behavioral attitudes, subjective norms, perceptual behavior control, and situational education are the main variables that determine behavioral intentions. The more positive the attitude, the greater the support of significant others, the stronger the perceived behavioral control, and the stronger the situational education intervention, the greater the behavior intention. Behavioral attitudes, subjective norms, perceptual behavior control, and situational environmental education are conceptually distinct, but sometimes they may share a common belief base, so they are both independent and related.

**Table 9 ijerph-19-11388-t009:** Path analysis of the structural model.

Path	Standard Error	Standardized Path Coefficient	Hypothesis
ATEB→REBI	0.032	0.660	Verified
SN→REBI	0.042	0.530	Verified
PBC→REBI	0.045	0.514	Verified
PBC→REB	0.043	0.535	Verified
REBI→REB	0.025	0.804	Verified
EE→REBI	0.029	0.733	Verified
EE→REB	0.026	0.776	Verified

Attitudes toward environmental behaviors (ATEB); responsible environmental behaviors (REB), responsible environmental behavioral intentions (REBI), situational environmental education (EE), perceptual behavior control (PBC), and subjective norm (SN).

**Figure 5 ijerph-19-11388-f005:**
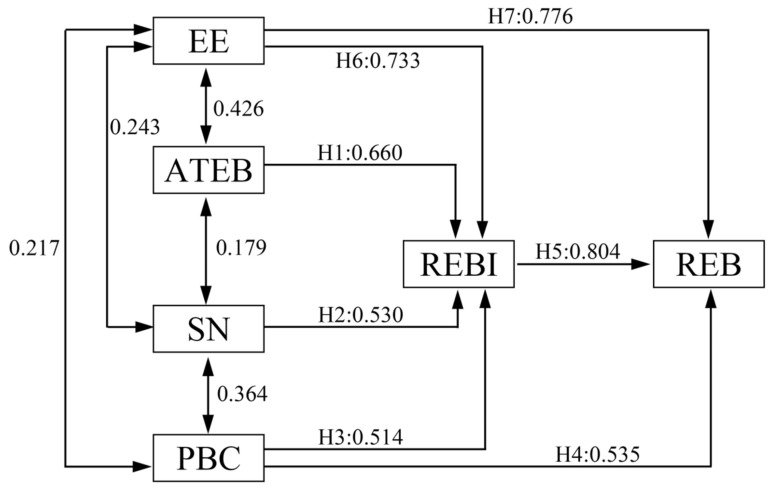
Results of the structural equation model. Attitudes toward environmental behaviors (ATEB), responsible environmental behaviors (REB), responsible environmental behavioral intentions (REBI), situational environmental education (EE), perceptual behavior control (PBC), and subjective norm (SN).

#### 5.2.4. Fitting Analysis of Situational Environmental Education Factors in the Model

By using the regression model between REB, REBI, and the intermediate variable EE, the fitting analysis was carried out to study the relationship between EE and REB and to measure the impact of EE on REB and REBI. Table 10 shows the test results. The result was R: 0.733, 0.776; R2: 0.537, 0.602; importance, 0.000, 0.000; B: 0.713, 0.731; and Beta, 0.733, 0.776, indicating that EE was highly positively correlated with REBI and REB, and the regression model fit well, which is statistically significant. Therefore, H6 and H7 were verified, and situational environmental education in tourist attractions had a positive influence on the tourists’ REBI and REB.

## 6. Discussion and Conclusions

In this study, we revealed a mental model of tourists’ responses to such changes based on a hierarchy from attributes to consequences and values, thus identifying the consequences of reactions to change, the reasons why these consequences occur, and their importance to tourists [79]. Related studies have shown that exploring tourists’ emotional experiences during tourism activities and psychological fluctuations play a crucial role in tourists’ behavior [16,80]. The study of behavior should be more of a combination of different methods and models [80,81]. The study is based on ZMET, involving in-depth interviews with 15 tourists to understand the factors that influence the occurrence of responsible environmental behavior in their tourist destinations. With this method, we can effectively avoid the uncertainty of the single questionnaire method, which likewise validates the previous study [5,82,83]. By investigating the most salient ladders that emerged in HVM, we found that situational environmental education can act as a covariate to influence tourists’ responsible environmental behavior intention and responsible environmental behavior. Situational environmental education shares the same factors as the original TPB model, both may share a common belief base, but are both independent and related [84].

This study analyzed the effect of situational environmental education about tourism destinations on tourists’ REB. By developing an EE-TPB model, the effect of environmental education on REB was studied and the following conclusions were drawn:Situational environmental education is a situational factor that influences tourists’ REB and has a significant positive impact on REBI and REB of tourists in tourist destinations.Situational environmental education as a covariate in the EE-TPB model may share a common belief base with the original TPB influences.Personal factors—including ATEB, SN, and PBC—have a strong positive effect on tourists’ REBI, which in turn has a very significant positive effect on tourists’ REB.The impact of educational activities related to experiential environmental education on tourists’ responsible environmental behavior is more evident in scenic environmental education.

Therefore, the author offers some suggestions for improvement to the destination:Tourists’ attitudes toward environmental behavior have a positive impact on their REBI. Therefore, the scenic beauty and cleanliness should be maintained so that tourists will consciously generate pro-environmental behavior. The services of scenic spots should be convenient for the people, with reasonable fees and standardized tours. The management should strengthen the advertising campaign to make tourists realize the importance and value of the scenic spots and guide their tourism consumption [85].Visitors’ SN has a positive impact on their REBI. Therefore, relevant departments should actively guide schools, communities, units, and other organizations to visit these places to create a positive environmental atmosphere and encourage tourists to regulate their REB.Visitors’ PBC has a positive impact on their REBI and REB. Therefore, the infrastructure of the scenic spot should be complete and well-maintained. The distribution of various signs and trash cans in the scenic area should be adequate and easy to use. Warning signs should be installed in places where uncivilized behavior is likely to occur to enhance the PBC of visitors [86].There should be some educational elements in the landscape, such as promoting the history of the landscape, implementing heritage conservation methods, providing environmental and resource knowledge [81], increasing signage, guiding visitors to adopt responsible environmental behavior, rewarding visitors who consciously engage in REB, and increasing compliance with regulations.Situational environmental education should be based on experiential environmental education, such as using 3D projection technology and virtual reality (VR) glasses to build environmental protection experience channels [87].

The study not only proposes a new method for studying tourists’ psychology, but also provides a scientific basis for cultivating tourists’ REB and provides feasible suggestions for creating scenic spots. It provides a scientific basis for relevant authorities to formulate effective, tourism-related environmental protection policies, conduct effective environmental management, identify environmental issues, and promote the sustainable development of these destinations.

The results of this study will help academia and the tourism industry. Future research could replicate the current study in other destinations to test the applicability of the data analysis and to compare findings across global tourism destinations [16]. Furthermore, subsequent studies should focus on the generality of the model as well as the influencing factors.

## Figures and Tables

**Figure 1 ijerph-19-11388-f001:**
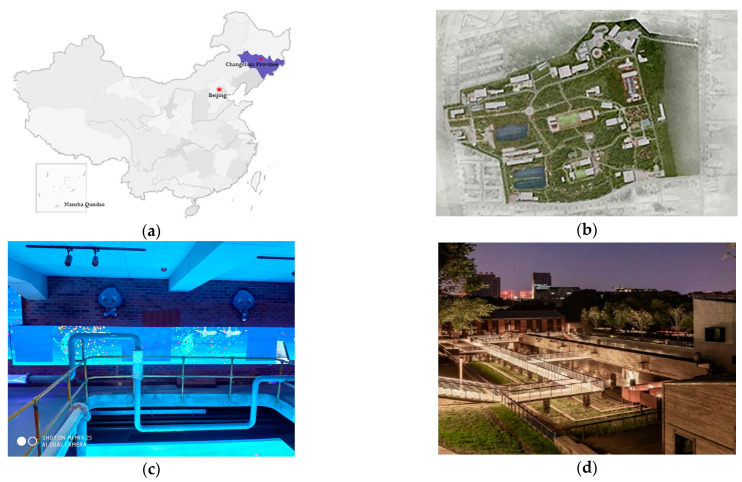
Maps and photos of the study area. (**a**) Location of Water Culture Ecological Park. (**b**) Aerial view of Water Culture Ecological Park. (**c**) Water purification flowchart. (**d**) Water purification plant sites.

**Figure 2 ijerph-19-11388-f002:**
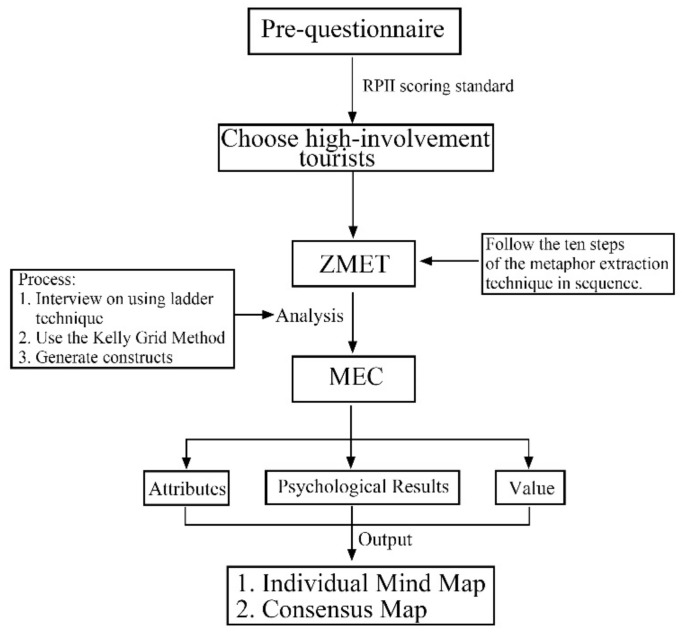
Flow chart of metaphor extraction technique.

**Figure 3 ijerph-19-11388-f003:**
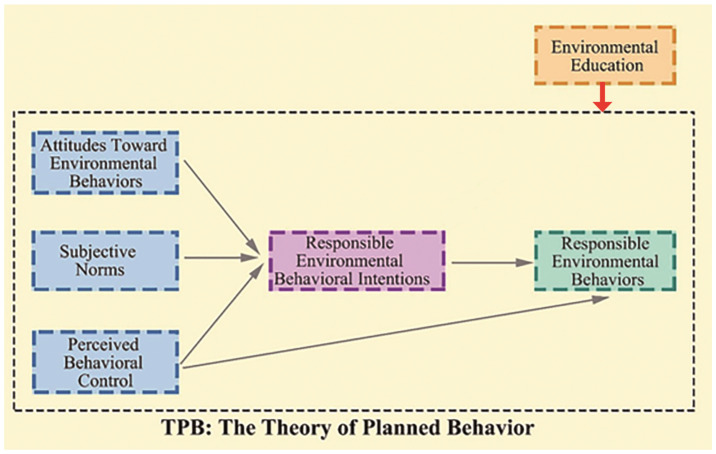
Structure of the relationship between situational environmental education and theory of planned behavior (TPB) model.

**Figure 4 ijerph-19-11388-f004:**
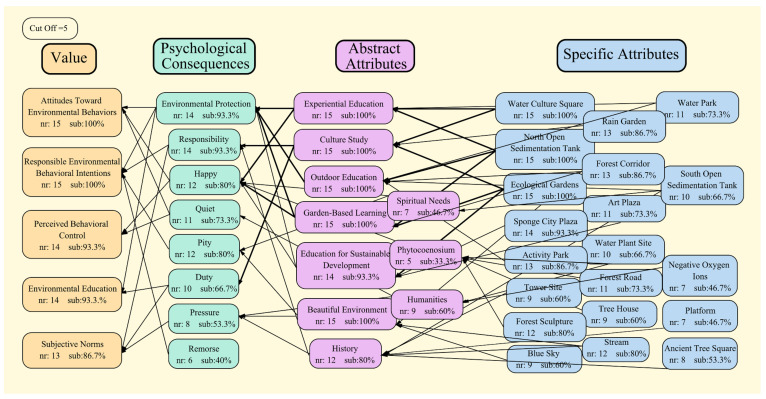
Consensus map developed using data from the 15 interviewees.

**Table 1 ijerph-19-11388-t001:** Data per the revised personal involvement inventory (RPII).

NO.	Gender	Hometown	Occupation	RPII Score
A	M	Changchun	Student	63
B	M	Changchun	Student	57
C	F	Ningbo	Student	58
D	M	Xian	Student	62
E	M	Dandong	Student	60
F	F	Shandong	Salesperson	58
G	F	Changchun	Salesperson	57
H	M	Changchun	Salesperson	64
I	M	Changchun	Profession	62
J	M	Changchun	Housewife	59
K	F	Chengde	Staff	63
L	M	Dalian	Staff	63
M	F	Shanghai	Staff	57
N	F	Changchun	Staff	62
O	M	Changchun	Staff	66

**Table 2 ijerph-19-11388-t002:** Zaltman metaphor elicitation technique (ZMET) procedure.

Interview Steps	Main Contents of Operation Method
Storytelling	Stories are an important foundation for human memory and communication. After more than a week of in-depth thinking on study topics, the respondents were asked to express their thoughts on the study topic by telling stories through pictures.
Missing Image	Interviewees were asked to describe the picture they were looking for but did not find and explain the special meaning of that picture. For the pictures that the interviewee did not find, guided by the ladder method, data were extracted by the concept of method-purpose chain.
Sorting Task	Interviewees were asked to classify all the collected pictures, based on their understanding, and to explain the reason for such classifications and the meaning of the representative image.
Construct Elicitation	Applied the Kelly grid method to obtain the deep and relevant answers from the interviewees and drew out the relationship between these answers through the ladder method. The two approaches complement each other and can effectively help interviewees express their ideas.
Most Representative Picture	Interviewees selected the most representative pictures related to the theme from the pictures they bring and explain why it was chosen. It enables the researcher to understand the inner thoughts of the interviewee more accurately.
Opposite Image	Interviewees were asked to find a picture that is contrary to the research topic or is a negative concept of the pictures provided. If not, asked the interviewees to describe what kind of a picture it should be.
Sensory Image	Sensory impressions help us understand the external world and reproduce it in memory. Asked the interviewees to use hearing, taste, touch, sight, smell, color, and emotion to describe what the research topic should and should not be.
Individual Mind Map	Extracted all the constructs mentioned by the interviewee about the tourist destination landscape; then, followed the interviewee’s constructive logic to integrate the typical construct. Finally, set up the interviewee’s mind map of the destination. Within a certain time, the consensus model was handed over to the interviewees to ensure the model was consistent with the idea being presented.
Summary Image and Vignette	Interviewees were asked to select pictures with which they could express their important ideas and provide a summary picture. This process can be described and pointed out by the interviewees and handled by the researcher.
Consensus Map	Combine the mind map of each interviewee to make a consensus map to show the principle of “Three Most”—most of the time, most of the people, and the most common ideas.

**Table 3 ijerph-19-11388-t003:** Constructs and scale used in the study.

Latent Variable	Observed Variable	Item Text	Reference
ATEB	ATEB1	I think it is wise to protect the scenic environment.	[42,43,44,48]
	ATEB2	I think it is necessary to protect the scenic environment.	
	ATEB3	I think it is valuable to protect the scenic environment.	
SN	SN1	People who are important to me think that I should protect the scenic environment.	[4,48]
	SN2	People I care about will blame me for not protecting the scenic environment.	
	SN3	The people I care about are protecting the scenic environment.	
PBC	PBC1	I think I have enough ability to protect the scenic environment.	[42,43,44,48]
	PBC2	I think I have enough ability to protect the scenic environment.	
	PBC3	Protecting the scenic environment is a happy thing.	
REBI	REBI1	I want to protect the scenic landscape from being destroyed.	[4,48]
	REBI2	I want to properly dispose of the garbage generated during travel.	
	REBI3	I do not want to buy products that affect the ecology of the scenic area.	
REB	REB1	I will protect the landscape of the scenic area from being damaged.	[4,48]
	REB2	I will properly dispose of the garbage generated during travel.	
	REB3	I will not buy products that damage the ecology of the scenic area.	
EE	EE1	Water culture education in the scenic area allows me to consciously regulate my REB.	
	EE2	The beautiful outdoor environment allows me to consciously regulate my REB.	
	EE3	Renewable energy education in the scenic area allows me to consciously regulate my REB.	

ATEB: attitudes toward environmental behaviors; SN: subjective norms; PBC: perceptual behavior control; REBI: responsible environmental behavioral intentions; REB: responsible environmental behaviors; EE: situational environmental education.

**Table 4 ijerph-19-11388-t004:** Hierarchical value map (HVM) developed in our study.

HVM	No.	Constructs	Number of Mentions
Specific Attributes	1	Water Culture Square	15
	2	North Open Sedimentation Tank	15
	3	Ecological Gardens	15
	4	Sponge City Plaza	14
	5	Activity Park	13
	6	Forest Corridor	13
	7	Rain Garden	13
	8	Stream	12
	9	Forest Sculpture	12
	10	Art Plaza	11
	11	Water Park	11
	12	Forest Road	11
	13	South Open Sedimentation Tank	10
	14	Water Plant Site	10
	15	Tree House	9
	16	Tower Site	9
	17	Blue Sky	9
	18	Ancient Tree Square	8
	19	Platform	7
	20	Negative Oxygen Ions	7
Abstract Attributes	1	Experiential Education	15
	2	Cultural Study	15
	3	Outdoor Education	15
	4	Garden-Based Learning	15
	5	Education for Sustainable Development	14
	6	Beautiful Environment	15
	7	History	12
	8	Humanities	9
	9	Spiritual Needs	7
	10	Phytocoenosium	5
Psychological Consequences	1	Environmental Protection	14
	2	Responsibility	14
	3	Pity	12
	4	Happy	12
	5	Quiet	11
	6	Duty	10
	7	Pressure	8
	8	Remorse	6
Value	1	Attitudes Toward Environmental Behaviors	15
	2	Responsible Environmental Behavioral Intentions	15
	3	Perceptual Behavior Control	14
	4	Environmental Education	14
	5	Subjective Norm	13

**Table 6 ijerph-19-11388-t006:** Reliability and convergent validity test results.

Variable	Mean	SD	Standardized Factor Loading	R^2^	CR	AVE	Cronbach’s Alpha
ATEB				0.667	0.946	0.854	0.914
ATEB1	4.58	0.388	0.935				
ATEB2	4.54	0.463	0.921				
ATEB3	4.58	0.443	0.917				
SN				0.508	0.841	0.638	0.747
SN1	4.20	0.720	0.827				
SN2	3.67	1.030	0.823				
SN3	4.14	0.752	0.744				
PBC				0.554	0.819	0.603	0.739
PBC1	3.88	0.904	0.843				
PBC2	3.40	1.199	0.786				
PBC3	4.29	0.598	0.693				
REBI				0.686	0.898	0.746	0.829
REBI1	4.29	0.580	0.882				
REBI2	4.36	0.538	0.877				
REBI3	4.25	0.663	0.831				
REB				0.611	0.906	0.762	0.843
REB1	4.30	0.567	0.877				
REB2	4.34	0.609	0.872				
REB3	4.25	0.658	0.870				
EE				0.766	0.909	0.770	0.850
EE1	4.32	0.602	0.897				
EE2	4.31	0.538	0.882				
EE3	4.30	0.751	0.853				

**Table 7 ijerph-19-11388-t007:** Analysis results of normality test.

	Kurtosis	Skewness	χ2	P
ATEB	−0.039	−0.507	1.334	0.452
SN	0.491	−0.606	5.586	0.061
PBC	0.040	−0.473	1.107	0.575
REBI	−0.510	0.032	4.217	0.121
REB	−0.256	−0.624	2.930	0.242
EE	0.138	−0.772	4.217	0.231

**Table 8 ijerph-19-11388-t008:** Discriminant validity test applied in our study.

Latent Variable	ATEB	SN	PBC	REBI	REB	EE
ATEB	0.924					
SN	0.480 **	0.917				
PBC	0.429 **	0.553 **	0.905			
REBI	0.660 **	0.530 **	0.514 **	0.948		
REB	0.656 **	0.541 **	0.525 **	0.804 **	0.952	
EE	0.599 **	0.535 **	0.497 **	0.733 **	0.776 **	0.953

Note: The diagonal includes the square root of the AVE; the correlations between the latent variables appear underneath the diagonal. ** *p* < 0.01. Attitudes toward environmental behaviors (ATEB), responsible environmental behaviors (REB), responsible environmental behavioral intentions (REBI), situational environmental education (EE), perceptual behavior control (PBC), subjective norm (SN).

**Table 10 ijerph-19-11388-t010:** Situational environmental education moderator variable analysis results.

Regression Analysis	R	R^2^	F	Significance	B	Standard Error	Beta
EE→REBI	0.733	0.537	608.232	0.000	0.713	0.029	0.733
EE→REB	0.776	0.602	792.515	0.000	0.731	0.026	0.776

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
