# Peer review of "Impact of Situational Environmental Education on Tourist Behavior—A Case Study of Water Culture Ecological Park in China"

_ijerph, 2022, doi:10.3390/ijerph191811388_

Round 1

Reviewer 1 Report

General comment:

Dear authors, thank you for the opportunity to read your work. the subject of responsible behaviour in tourism is an important one and one that needs more work. I find your methodological approach interesting. However, my main concern is that your research objectives are not clear (and sometimes are not research objectives) and that the hypotheses you propose (and therefore the results that follow) do not add much to the theory you are mobilising. To be more precise, your hypotheses concern the relationships between the dimensions of the theory of planned behaviour and the responsible environmental behaviour intentions, but those relations are already known and constitute the theoretical basis of TPB. However, the environmental education aspect of your manuscript is interesting and could be a basis for your revision.

Below are more specific comments that I hope will help you to revise your manuscript and increase its quality.

Introduction

In general, I find that the introduction is too broad on the concepts covered and at the same time too short on the content of each of them. In view of the concepts mobilised, I feel that an additional section should be created concerning the theories/concepts (A litterature review on MEC, TPB, etc.) and that the introduction should focus on the justification of this work and its place in current issues (which does not prevent a few words being said about the theories used).

Paragraph p2. L81-90 seems to me to be more a justification of the method than a problematic. Its place in a reorganisation should perhaps be reviewed.

It also provides a justification of the theory used. Why the TPB and not the VBN for example? In the same way, you bring the MEC theory in an abrupt way and one does not understand why this one rather than another. Again, I think an intro/literature review reorganization is to be considered

The purpose section could be joint to the intro without being a specific one.

That said, there is a problem with your objectives:

(1) guiding visitors to express something does not seem to me to be a research objective but rather an objective of a method in order to answer a research question.

(2) Yes, understanding the influence of situational environmental education on LERs is a research objective.

(3) Unless I am mistaken, this is not what you do in the manuscript. However, you propose options for stakeholders to do so

(4) this is a methodological aspect to reduce bias.

I think you need to revise your research objectives. Ask yourself: what are my research questions, what is the purpose of this article in terms of adding to the existing body of knowledge.

I failed to understand what SITUATIONAL environmental education is compare to environmental education

Research hypotheses

Why focusing on responsible env intention and not on pro-environmental behaviour intentions? There is literature on this point, for eg related to polar beer or glacier that I think could be useful in your study. You may justify why REBI and not PEBI

In Figure 1, you show a potential relationship between environmental education and REBI and REB. However, aren't environmental attitudes, subjective norms and perceived behavioural control antecedents of these elements? Thus, isn't environmental education more related to these elements of TPB?

The hypotheses you propose are in my opinion already known. The theory of planned behaviour predicts that intentions, and by extension behaviours, have as antecedents attitudes, subjective norms and PBC. Your hypotheses are therefore already validated by the theory.

See for eg this book: https://onlinelibrary.wiley.com/doi/book/10.1002/9781119241072

On the other hand, and I have the impression that this is the interest of your article, does environmental education have an influence on the three elements of TPB? and therefore, by extension, on REBI and REB?

Data, sources and research methods

Add the country in the first sentence (we know it, but I think it more convenient for the quick reader)

When were the interviews conducted? What are the differences between interview and pre-interview?

The methodology is quite unclear. I think you have a very interesting methodology but the way it is presented makes it unclear. For example, after the presentation of the study site and before point 3.2, I think there should be a paragraph summarising the logical sequence of the method. The presentation of the case study could also come after this presentation.

In table 2, some concepts are used without being defined: this is the case for the purpose-chain and the Kelly grid method which are neither associated with a reference, nor clearly defined.

Analytical methodology

4.1: How were the questionnaires collected, by whom? Be more specific

I think that helsinski's statement should be sourced and explained in a few words and how this questionnaire respects it.

Results

I am not familiar with the ZMET technique. What are the steps, structures and connections you are talking about? Try to explain this better

In relation to my previous comment, you should include EE in your structural equation (if possible). It would be interesting to know if EE influences the antecedents of intentions. You show that EE influences REBI and REB, but does the relationship not go through the antecedents as suggested by TPB?

Discussion

My main comment here is that your points 1 to 3 do not add much to the research on pro-environmental behaviour. You show very well that the antecedents of intentions have an influence on intentions but this is already the basis of TPB.

However, your contribution could be related to the influence of EE on antecedents and the place of EE in these tourist sites.

The discussion should be reviewed in view of the changes in the other sections.

In addition, you do not mobilise the literature sufficiently. There are studies that exist on pro-environmental behaviour that would be very useful to discuss. I'm thinking in particular of work on last chance tourismand PEB, or Halpenny's work on national parks. You need to discuss your results in the light of these other studies.

It is strange to have a conclusions section before the main conclusions…

In any cases, it will require revision in relation to the changes made in other sections.

Reviewer 2 Report

The article discusses a hot topic in tourism research, i.e., tourist environmentally responsible behavior (TERB). I suggest the following should be emphasized or modified:

  1. The considerations for introducing tourist situational environmental education and its value need to be elaborated
  2. As for the Theory of Planned Behavior, the importance and applicability of TPB and its extension could be enriched;
  3. As to 2.3. Perceptual behavior control, why does PBC has direct and positive influences on both intention and actual behavior, while attitudes do not have the dual influences? It should be explained more.
  4. The typicality of the case--Changchun Water Culture Ecological Park is not well represented. How about its honor(s) and annual visit number?
  5. The normal distribution test, common method variance test, and mediation effect test were missing;
  6. Sessions 6.2 and 7 have the same title, which is not appropriate for “conclusions” and both should be renamed;
  7. There are many authoritative references in the field of TERB, however, the overall references of this article are lagging behind. Authors can refer to some recent articles. In 2022, for instance, related papers include “The effect of destination employee service quality on tourist environmentally responsible behavior: A moderated mediation model incorporating environmental commitment, destination social responsibility and motive attributions” (He et al.), “The effect of destination source credibility on tourist environmentally responsible behavior: an application of stimulus-organism-response theory” (Qiu et al., https://doi.org/10.1080/09669582.2022.2067167), “Landscape and Unique Fascination: A Dual-Case Study on the Antecedents of Tourist Pro-Environmental Behavioral Intentions” (Zheng et al.), “Past, present, and future of pro-environmental behavior in tourism and hospitality: a text-mining approach” (Loureiro et al.), “Urban travelers’ pro-environmental behaviors: Composition and role of pro-environmental contextual force” (Qin & Hsu).
  8. Theoretical contributions and practical implications can be polished further.

Round 2

Reviewer 1 Report

Thank you very much for this revised version. The quality of the paper has been greatly improved. Here are just some minor comments to be adress before publication :

- You use "NAT" but did not defined it neither explaining what is the signification of the acronym

- In the discussion, you propose that virtual reality can be a tool for environmental education. Can you add a reference that justifies this statement.

- A quick discussion, using literature, of each of your suggestions would also be a plus.

Reviewer 2 Report

It’ admitted that this manuscript quality has been improved significantly via the authors’ revision. In this sense, this paper has been close to accept for IJERPH. However, there are several minors that need to further revise.

(1) In data collection section, the time of pre-questionnaire and interview is omitted as to which year.

(2) In reference section, authors need to pay attention to the mistake made by repeating the same reference of Zheng et al. 2022. The 86th literature should be deleted, however, the 62th literature should be retained. Additionally, the similar mistakes should be revised in another round revision.

(3) The potential important and new literature on tourist environmentally responsible behavior/tourist pro-environmental behavior has been published in 2022. The following literature is encouraged to add it to reflect the timeliness of this research in the revision. For example,

[1] Qiu, H.; Wang, X.; Wu, M.-Y.; Wei, W.; Morrison, A.M.; Kelly, C. The Effect of Destination Source Credibility on Tourist Environmentally Responsible Behavior: An Application of Stimulus-Organism-Response Theory. J. Sustain. Tour. 2022, 1-21. https://doi.org/10.1080/09669582.2022.2067167

[2] Qiu, H.; Wang, X.; Morrison, A.M.; Kelly, C.; Wei, W. From Ownership to Responsibility: Extending the Theory of Planned Behavior to Predict Tourist Environmentally Responsible Behavioral Intentions. J. Sustain. Tour. 2022, 1-24. https://doi.org/10.1080/09669582.2022.2116643

(4) the capacity English writing is very crucial to any academic paper. Here, an native-English editor is encourage to proofread in the following revision.

Very close to the goal!

Keep on fighting!

Good Luck!
